# ERVK13-1/miR-873-5p/GNMT Axis Promotes Metastatic Potential in Human Bladder Cancer though Sarcosine Production

**DOI:** 10.3390/ijms242216367

**Published:** 2023-11-15

**Authors:** Shingo Kishi, Shiori Mori, Rina Fujiwara-Tani, Ruiko Ogata, Rika Sasaki, Ayaka Ikemoto, Kei Goto, Takamitsu Sasaki, Makito Miyake, Satoru Sasagawa, Masashi Kawaichi, Yi Luo, Ujjal Kumar Bhawal, Kiyohide Fujimoto, Hidemitsu Nakagawa, Hiroki Kuniyasu

**Affiliations:** 1Department of Molecular Pathology, Nara Medical University, 840 Shijo-cho, Kashihara 634-8521, Japan; nmu6429@yahoo.co.jp (S.K.); m.0310.s.h5@gmail.com (S.M.); rina_fuji@naramed-u.ac.jp (R.F.-T.); pkuma.og824@gmail.com (R.O.); rika0st1113v726296v@icloud.com (R.S.); a.ikemoto.0916@gmail.com (A.I.); ilgfgtk@gmail.com (K.G.); takamitu@fc4.so-net.ne.jp (T.S.); 2Research Institute, Tokushukai Nozaki Hospital, 2-10-50 Tanigawa, Daito 574-0074, Japan; satoru.sasagawa@tokushukai.jp (S.S.); masashi.kawaichi@tokushukai.jp (M.K.); nakagawa.hide@tokushukai.jp (H.N.); 3Department of Urology, Nara Medical University, Kashihara 634-8522, Japan; makitomiyake@yahoo.co.jp (M.M.); kiyokun@naramed-u.ac.jp (K.F.); 4Jiangsu Province Key Laboratory of Neuroregeneration, Nantong University, 19 Qixiu Road, Nantong 226001, China; lynantong@hotmail.com; 5Research Institute of Oral Science, Nihon University School of Dentistry at Matsudo, Matsudo 271-8587, Japan; bhawal2002@yahoo.co.in; 6Center for Global Health Research, Saveetha Medical College and Hospitals, Saveetha Institute of Medical and Technical Sciences, Saveetha University, Chennai 600077, India

**Keywords:** SAM, GNMT, sarcosine, cancer metastasis, bladder cancer

## Abstract

N-methyl-glycine (sarcosine) is known to promote metastatic potential in some cancers; however, its effects on bladder cancer are unclear. T24 cells derived from invasive cancer highly expressed GNMT, and S-adenosyl methionine (SAM) treatment increased sarcosine production, promoting proliferation, invasion, anti-apoptotic survival, sphere formation, and drug resistance. In contrast, RT4 cells derived from non-invasive cancers expressed low GNMT, and SAM treatment did not produce sarcosine and did not promote malignant phenotypes. In T24 cells, the expression of miR-873-5p, which suppresses GNMT expression, was suppressed, and the expression of ERVK13-1, which sponges miR-873-5p, was increased. The growth of subcutaneous tumors, lung metastasis, and intratumoral GNMT expression in SAM-treated nude mice was suppressed in T24 cells with ERVK13-1 knockdown but promoted in RT4 cells treated with miR-873-5p inhibitor. An increase in mouse urinary sarcosine levels was observed to correlate with tumor weight. Immunostaining of 86 human bladder cancer cases showed that GNMT expression was higher in cases with muscle invasion and metastasis. Additionally, urinary sarcosine concentrations increased in cases of muscle invasion. Notably, urinary sarcosine concentration may serve as a marker for muscle invasion in bladder cancer; however, further investigation is necessitated.

## 1. Introduction

Bladder urothelial cancer (BUC) is estimated to affect 168,560 people and cause 32,590 deaths in the United States in 2023 [1]. In Japan, 23,383 people were affected, and 9168 died in 2020 [2]. Approximately 80% of newly diagnosed BUC cases are non-muscle invasive and are treated with transurethral resection. Tumor recurrence occurs in 70% of treated cases, and progression to invasive disease occurs in 15% of the cases [3]. In contrast, muscle-invasive bladder carcinoma (MIBC) is more aggressive, with a 50% mortality rate due to dissemination [4]. In these cases, more aggressive therapy with radical cystectomy and urinary diversion or trimodal therapy with maximal endoscopic resection, radiosensitizing chemotherapy, and radiation is implemented, and immune checkpoint inhibitors and molecular-target drugs are used for recurrence and metastasis [5]. Therefore, to treat BUC, there is a need to develop new effective markers and therapeutic agents for MIBC.

The causes of malignant phenotypes in MIBC include invasive ability, metastatic ability, drug resistance due to the induction of epithelial–mesenchymal transition (EMT) [6], and enhancement of stemness [7]. We previously reported that maintenance of the tumor microenvironment by tight junctions plays a vital role in such malignant phenotypes [8]. Epigenetic modifications to gene-coding regions have been known to affect their expression in MIBC. In MIBC, claudin-4 expression is enhanced by the hypomethylation of its gene promoter [9].

In contrast, the epigenetic regulation of gene expression by microRNAs (miRNAs) and long non-coding RNAs (lncRNAs) has attracted attention because it influences the malignant phenotypes of cancer [10]. Long non-coding RNAs (lncRNAs) are oligonucleotides of more than 200 nucleotides in length that have no protein-coding potential [11]. lncRNAs suppress the action of miRNAs by sponging them, adding complexity to the regulation of gene expression [12]. We previously revealed that HOXA11-AS sponges miR-494 and regulates cancer-related gene expression in oral cancers [13]. MiRNAs and lncRNAs are thought to play roles in the development of bladder cancer [14], and UCA1, HOTAIR, and GAS5 are potential markers for the diagnosis and prognosis of bladder cancer [15].

Sarcosine (N-methylglycine) has attracted attention as one of the oncometabolites [16] and is synthesized from glycine by glycine-N-methyltransferase (GNMT), with S-adenosyl-methionine (SAM) serving as the methyl donor in this process [17]. In contrast, it is metabolized to glycine by sarcosine dehydrogenase. Sarcosine induces autophagy, which decreases with aging [18]. Sarcosine is excreted in urine and is expected to be a diagnostic marker for prostate cancer [19]. Due to its association with metastasis, sarcosine has been shown to be a promoter of cancer malignancy in prostate, kidney, and breast cancers [17,20]. However, the relationship between malignancy and urothelial cancer remains unclear.

In this study, we analyzed the relationship between sarcosine and malignant phenotypes of BUC.

## 2. Results

### 2.1. Effect of SAM on BUC Cell Malignancy 

To examine the effects of GNMT, we treated BUC cells with SAM, which serves as a methyl group donor for GNMT activity (Figure 1). When the human BUC cell lines T24 and RT4 were treated with SAM, proliferation was promoted in a concentration-dependent manner in T24 cells but not in RT4 cells (Figure 1A). Furthermore, enhanced invasive ability was observed in T24 cells after SAM treatment but not in RT4 cells (Figure 1B). Additionally, SAM also promoted sphere formation, but only in the T24 cells (Figure 1C,D).

### 2.2. Effect of Sarcosine on BUC Cell Malignancy 

Next, we investigated the effects of sarcosine produced by GNMT (Figure 2). In T24 cells, SAM administration increased sarcosine concentration in a dose-dependent manner, but this effect was not observed in RT4 cells (Figure 2A). Treatment of both cell types with sarcosine promoted proliferation, invasion, anti-apoptotic survival, drug resistance to CDDP, and sphere formation (Figure 2B–F). Expression of the stem cell markers OCT3 and CD44 was promoted by SAM in T24 cells but not in RT4 cells. In contrast, sarcosine treatment promoted this effect in both cell lines (Figure 2G).

### 2.3. Effect of GNMT on BUC Cell Malignancy

The expression of GNMT, which converts SAM to sarcosine, was 5.6 times higher in T24 cells than in RT4 cells (Figure 3A). In contrast, the expression of sarcosine dehydrogenase (SADH), a sarcosine-degrading enzyme, was 1.6 times higher in RT4 cells than in T24 cells. When GNMT expression was knocked down in T24 cells, the effects of SAM on promoting proliferation, suppressing apoptosis, and promoting sphere formation were abrogated (Figure 3B–E).

**Figure 2 ijms-24-16367-f002:**
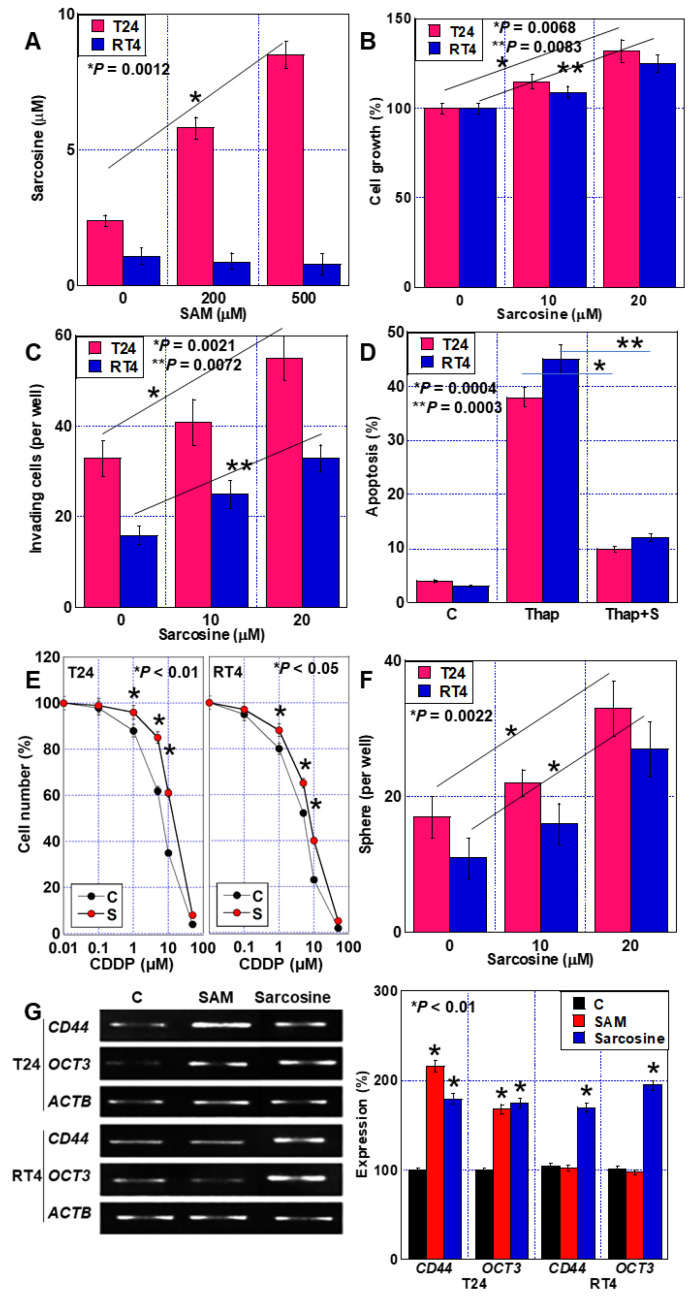
Effect of sarcosine on human BUC cell malignancy. (**A**) Intracellular sarcosine concentrations in SAM-treated BUC cells. (**B**–**G**) Effect of sarcosine on cell growth (**B**), invasion (**C**), thapsigargin-induced apoptosis (**D**), drug resistance to CDDP (**E**), sphere formation (**F**), and expression of stemness-associated genes (**G**) with semi-quantification (right panel). Error bars: standard deviation from three independent trials. Statistical significance was calculated using the two-way ANOVA test or the Student’s *t*-test. The assumption test suggests that the difference between the two SDs is not significant. BUC, bladder urothelial carcinoma; SAM, S-adenosylmethionine; Thap, thapsigargin; S, SAM; C, control; CDDP, cisplatin; OCT3, octamer-binding transcription factor; ACTB, β-actin.

### 2.4. Epigenetic Regulation of GNMT Expression by miRNAs and lncRNAs

The expression of miR-873-5p, which suppresses the expression of GNMT, was found to be 1/4 of that of RT4 cells in T24 cells (Figure 4A). Subsequently, when T24 cells were treated with a miR-873-5p mimic, GNMT expression was suppressed, and SAM-induced proliferation and sphere formation were inhibited (Figure 4B–D). In contrast, miR-873-5p inhibitor treatment increased GNMT expression and promoted cell proliferation and sphere formation. However, when RT4 cells were treated with the miR-873-5p mimic, GNMT expression was almost completely abolished, and SAM-induced proliferation and sphere formation remained suppressed. Furthermore, treatment with the miR-873-5p inhibitor promoted GNMT expression, SAM-induced proliferation, and sphere formation.

Additionally, we examined the expression of three lncRNAs reported to regulate miR-873-5p, namely, ERVK13-1 [21], MALAT1 [22], and DDX11-AS1 [23,24]. The expression of ERVK13-1 was 5 times higher in T24 cells than in RT4 cells. (Figure 4E), whereas MALAT1 and DDX11-AS1 expression were not detected in any of the cells. Further, ERVK13-1 knockdown abolished the proliferation and sphere-formation-promoting effect of SAM in T24 cells (Figure 4F).

### 2.5. Effects of Altered GNMT Expression on Tumor Growth and Metastasis in BUC Cells

We epigenetically altered GNMT expression and examined its effects in a mouse model (Figure 5). T24 cells, T24 cells with ERVK13-1 knockdown, RT4 cells, and RT4 cells treated with miR-873-5p inhibitor were inoculated subcutaneously (Figure 5A). ERVK13-1 knockdown in T24 cells reduced tumor growth by 1/4. Contrastingly, in RT4 cells, tumor growth increased 1.3-fold upon miR-873-5p suppression. Next, the cells were inoculated into the caudal vein to examine lung metastases (Figure 5B). Lung metastasis by T24 cells was reduced by 83.3% by knockdown of ERVK13-1. Meanwhile, miR873-5p inhibition in RT4 cells promoted lung metastasis by 1.5 fold. In subcutaneous tumors, GNMT expression decreased to 45% in T24 cells due to ERVK13-1 knockdown and increased by 2.5-fold in RT4 cells due to miR-873-5p suppression (Figure 5C). Furthermore, when urinary sarcosine was measured at the time of euthanasia in the subcutaneous tumor model, it was reduced to 50% by ERVK13-1 knockdown in T24 cells and increased by 1.7-fold in RT4 cells by miR-873-5p suppression (Figure 5D). In contrast, in the lung metastasis model, ERVK13-1 knockdown in T24 cells decreased it to 20%, and miR873-5p inhibitor treatment in RT4 cells increased it 3.5-fold. Thus, GNMT expression may be associated with cancer growth and metastasis, and it is possible to predict GNMT expression using its product, urinary sarcosine.

### 2.6. Significance of GNMT Expression in Human BUC Cases

Finally, to evaluate the role of GNMT in human BUC, we examined GNMT expression in 86 human BUC cases by immunostaining (Table 1, Figure 6). Immunostaining showed that GNMT was immunoreactive in the cytoplasm of cancer cells with slight intratumoral heterogeneity (Figure 6A). In pTa/pTis-pT2 cases, GNMT positivity was 15%, whereas in pT3-4 cases, the positivity was 81%; in distant metastasis-positive cases (M1b cases), the positivity was significantly high. When comparing non-muscle invasion (NMIBC) and muscle-invasive (MIBC) cases, significantly more GNMT-positive cases were observed in MIBC. Furthermore, in cases in which urine could be analyzed immediately before surgery, urinary sarcosine levels increased as the T factor (primary tumor invasion) progressed (Figure 6B). Additionally, these levels were significantly higher in patients with MIBC than in those with NMIBC (Figure 6C).

## 3. Discussion

In this study, we found that sarcosine levels were strongly associated with malignant phenotypes in patients with BUC. Notably, the lncRNA ERVK13-1 promotes the expression of GNMT by sponging miR-873-5p, which suppresses the expression of sarcosine-synthesizing GNMT. When investigating the effect of sarcosine on BUC, we proceeded with the research as shown in Figure 7A. The role of the effector SAM, sarcosine, and sarcosine synthase GNMT, the regulation of GNMT expression by epigenetics via microRNA and lncRNA, validation of these results using mouse models, and using human BUC cases. As a result of these studies, the tumor-promoting effect of sarcosine on BUC has a consistent flow of the expression of its production enzyme and the regulation system of the expression of the enzyme, and that ERVK13-1/miR-873-5p/GNMT axis, which is essential for the expression of the action of sarcosine.

Our data showed that external administration of sarcosine enhanced stemness in both T24 cells, which overexpress GNMT, and RT4 cells, which have lower GNMT. Stemness is a factor that causes cancer to develop malignant phenotypes, such as metastasis and drug resistance [26]. This is thought to be the basis for the mechanism of stemness induction by sarcosine. However, the precise mechanism has not yet been clarified, although a relationship with zinc accumulation has been reported [27]. Additionally, GNMT maintains an optimal ratio of SAM to S-adenosylhomocysteine (SAH), regulates the supply and utilization of methyl groups, and influences homocysteine balance [28]. In liver carcinogenesis, GNMT mutations cause an imbalance in the SAM:SAH ratio, which subsequently increases the risk of carcinogenesis via aberrant methylation [29]. However, chronic excess of SAM due to GNMT inactivation also induces carcinogenesis [30]. This suggests that in addition to the direct action of sarcosine, the imbalance in the SAM: SAH ratio, caused by GNMT activation, might lead to enhanced malignant phenotypes.

In this study, we found that increased GNMT expression strongly correlated with the malignancy of BUC. Suppression of GNMT expression by CpG promoter hypermethylation and activation of the PI3K pathway have been reported [31,32]. In recent years, the epigenetic regulation of gene expression by miRNAs has become clear. lncRNAs sponge miRNAs, abolishing their effects, thereby regulating gene expression together with miRNAs and attracting the attention of several researchers [33]. miR-873-5p, which was investigated in this study, suppresses GNMT expression in liver cirrhosis [34] and non-alcoholic steatohepatitis [35]. It also promotes the progression of liver cirrhosis.

Our data also revealed that ERVK13-1 promotes GNMT expression by sponging miR-873-5p. Regarding the effect of lncRNA on miRNA, we have also reported that sponging of miRNA-494 by HOXA11-AS enhances the expression of nicotinamide adenine dinucleotide (NAD)(P)H: quinone oxidoreductase 1 [13]. ERVK13-1 suppresses miR-873-5p by sponging it, promoting the expression of its target gene KLF5 and enhancing malignant phenotypes in osteosarcoma [21]. In our study, KLF5 expression was not altered by miR-873-5p mimics and inhibitors or ERVK13-1 knockdown in BUC cell lines.

Sarcosine is excreted in the urine as prostate cancer progresses and is, therefore, a potential diagnostic marker [18]. In BUC, the presence or absence of muscle invasion affects treatment selection and prognosis, and although this preoperative diagnosis is required, no reliable marker has been established to date. We found that GNMT expression was correlated with BUC progression, particularly metastasis. Additionally, we showed that urinary sarcosine concentration could distinguish MIBC from non-MIBC cases. This suggests that MIBC can be diagnosed preoperatively using a simple and clinically useful method of urine testing. Unfortunately, in our study, measurements of urinary sarcopenia remained in a small panel. To obtain greater reliability, analysis of a large number of prospective cases is necessary. The confidence in using sarcosine as a biomarker can be improved by extensive analysis of more samples.

Our study shows that epigenetic regulation by miRNAs and lncRNAs induces GNMT expression and promotes sarcosine production, thereby promoting muscle invasion and distant metastasis in BUC. This demonstrates a novel target for the diagnosis and treatment of BUC; however, further detailed clinical investigations are necessitated.

## 4. Materials and Methods

### 4.1. Cells and Reagents

T24 and RT4 human BUC cell lines were purchased from the American Type Culture Collection (ATCC; Manassas, VA, USA). Cells were cultured in Dulbecco’s modified Eagle’s medium (Wako Pure Chemical Corporation, Osaka, Japan) supplemented with 10% fetal bovine serum (FBS; Sigma, St. Louis, MO, USA). Cell growth was assessed using the 3-(4,5-dimethylthiazol-2-yl)-5-(3-carboxymethoxyphenyl)-2-(4-sulfophenyl)-2H-tetrazolium (MTS)-based CellTiter 96 Aqueous One Solution Cell Proliferation Assay kit (Promega Biosciences Inc., San Louis Obispo, CA, USA), as previously described [36]. Apoptosis was induced by 5 μM thapsigargin (Wako Pure Chemical Corporation, Osaka, Japan), as previously described [37]. Briefly, a total of 1000 cells were stained with Hoechst 33342 dye (Life Technologies, Carlsbad, CA, USA) and examined under a fluorescence microscope. The invasion was assessed using a modified Boyden chamber assay to examine the in vitro invasion ability of BUC cells [36]. After 24 h of incubation at 37 °C, the filter was carefully removed from the inserts, stained with hematoxylin for 10 min, and mounted onto microscopic slides. The number of stained cells in the entire insert was counted at 100× magnification. Invasion activity was quantified as the average number of cells per well. All experiments were performed in triplicates. Cells were treated with cisplatin (CDDP, WAKO), SAM (Sigma), sarcosine (Sigma), and *miR-873* mimic and inhibitor (Invitrogen, Carlsbad, CA, USA) for 48 h. All experiments were performed in triplicate. 

### 4.2. Sphere Formation Assay

Cells (1000 cells/well) were seeded in uncoated bacteriological 35-mm dishes (Corning Inc., Corning, NY, USA) with 3D Tumorsphere Medium XF (Sigma). After culturing for 7 days, sphere images taken by an inverted microscope coupled with a camera were captured on a computer, and the sphere number was measured using NIH ImageJ software (version 1.52, Bethesda, MD, USA).

### 4.3. Reverse Transcription–Polymerase Chain Reaction (RT–PCR)

To assess human mRNA expression, RT–PCR was performed with 0.5 µg total RNA extracted from the three cell lines using the RNeasy kit (Qiagen, Germantown, MD, USA). The primer sets used are listed in Table 2 and were synthesized by Sigma Genosys (St. Louis, MO, USA). The PCR products were electrophoresed on a 2% agarose gel and stained with ethidium bromide. *ACTB* mRNA was amplified and used as the internal control. 

### 4.4. Quantitative Reverse Transcription-Polymerase Chain Reaction 

Extraction of total RNA was carried out using an RNeasy Mini Kit (Qiagen), and total RNA (1 μg) was synthesized using the ReverTra Ace-α-RT Kit (Toyobo, Osaka, Japan). Quantitative reverse transcription-polymerase chain reaction (qRT-PCR) was performed by using StepOne Real-Time PCR System with Fast SYBR^®^ Green Master Mix (Applied Biosystems, Life Technologies, Carlsbad, CA, USA) and a relative standard curve quantification method was used for analysis [39]. PCR was performed according to the manufacturer’s instructions. *ACTB* mRNA was used as the internal control. Each amplification reaction was evaluated using melting curve analysis. The PCR products were visualized by agarose gel electrophoresis and ethidium bromide staining.

### 4.5. Small Interfering RNA

Stealth selects RNAi interference (siRNA) targeting human *GNMT* and *ERVK113-1* was purchased from Sigma-Aldrich. AllStars Negative Control siRNA was used as a control (Qiagen, Valencia, CA, USA). Cells were transfected with 10 nM siRNA using Lipofectamine 3000 (Thermo Fisher Scientific, Tokyo, Japan) according to the manufacturer’s recommendations.

### 4.6. Mouse Models

Four-week-old male Slc-nu/nu BALB/c mice were purchased from SLC Japan (Shizuoka, Japan). The initial weight of the mouse was 19 ± 0.4 g. Mice were maintained in accordance with the institutional guidelines approved by the Committee for Animal Experimentation of Nara Medical University and the current regulations and standards established by the Ministry of Health, Labor, and Welfare (approval number 12733, 14 February 2018). The mice were administered SAM in drinking water (1.0 mg/mL) via free drinking (the mean intake of SAM was 6.6 mg/mouse/day), which corresponds to 20 mg/kgBW/day [40]. 

The pCMV6-A-GFP Mammalian Expression Vector (OriGene Technologies, Rockville, MD, USA) was transfected into T24 and RT4 cells using Lipo3000 (Thermo Fisher Scientific) according to the manufacturer, and Welfare (approval number 12733, 14 February 2018). The micion with siERVK13-1 (Sigma) or siControl (Qiagen) for 48 h. RT4 cells were pretreated before inoculation with the miR-873 inhibitor (Invitrogen) for 48 h. 

To establish a subcutaneous tumor model, BUC cells (1 × 10^7^) were inoculated into the subcutaneous tissues of nude mice. For the lung metastasis model, BUC cells (1 × 10^6^) were inoculated into the caudal vein. To assess tumor volume by GPT fluorescence intensity, the mice were examined weekly using a Clairvivo OPT in vivo imager (Shimazu, Kyoto, Japan) under anesthesia. Immediately before euthanasia, the lower abdomen of the mice was compressed for urination, and urine was collected. After euthanasia under inhalation anesthesia with 3% isoflurane, subcutaneous tumors were collected, and RNA was extracted using the TRIzol method [41]. 

### 4.7. Patients

Eighty-six patients with bladder cancer who were treated with transurethral resection or total cystectomy at Nara Medical University Hospital were randomly selected. The basic patient information is summarized in Table 1. Since written informed consent was not obtained, any identifying information was removed from the samples prior to analysis to ensure strict privacy protection (unlinkable anonymization). All procedures were performed in accordance with the Ethical Guidelines for Human Genome/Gene Research enacted by the Japanese Government and approved by the Ethics Committee of Nara Medical University (Approval Number 937, 10 October 2010).

### 4.8. Immunohistochemistry 

Consecutive 4 μm sections were immunohistochemically stained using the immunoperoxidase technique described previously. The anti-GNMT antibody (Abcam, Cambridge, UK) was used at a concentration of 0.2 µg/mL, and secondary antibodies (Medical and Biological Laboratories, Nagoya, Japan) were used at a concentration of 0.2 µg/mL. Tissue sections were color-developed using diaminobenzidine hydrochloride (DAKO, Glastrup, Denmark) and counterstained with Meyer’s hematoxylin (Sigma-Aldrich). The cells with immunoreactions in the cytoplasm were counted. The staining grade was scored from 0 to 3. 

### 4.9. Statistical Analysis

Statistical significance was calculated using two-tailed Chi-square, two-way ANOVA, and unpaired Mann–Whitney tests using InStat software ver. 3.10 (GraphPad, Los Angeles, CA, USA). Statistical significance was defined as a two-sided *p*-value < 0.05.

## 5. Conclusions

As shown in Figure 7B, sarcosine is produced by GNMT and promotes the malignancy of BUC. GNMT expression is epigenetically regulated by miR-873-5p, a suppressor, and ERVK13-1, a promoter. Sarcosine is excreted in the urine, and its concentration may predict MIBC and distant metastasis. Thus, sarcosine and its production system are expected to be a new therapeutic target for BUC. 

## Figures and Tables

**Figure 1 ijms-24-16367-f001:**
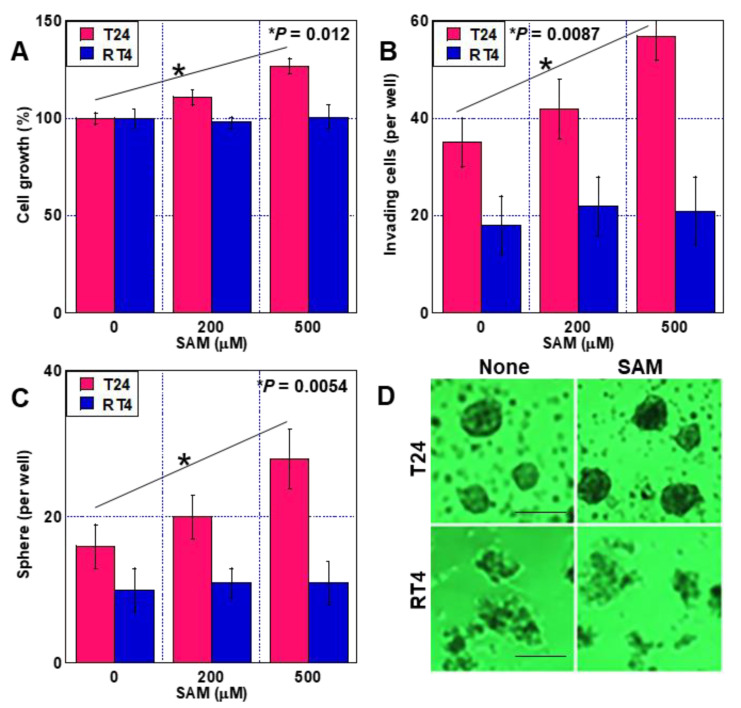
Effect of SAM on BUC cell malignancy. (**A**–**C**) The effect of SAM on cell growth (**A**), invasion (**B**), and sphere formation (**C**) as examined in human BUC, T24, and RT4 cells. (**D**) Microscopic image of the sphere. Scale bar, 50 μm. Error bars: standard deviation from three independent trials. Statistical significance was calculated using two-way ANOVA. BUC, bladder urothelial carcinoma; SAM, S-adenosylmethionine.

**Figure 3 ijms-24-16367-f003:**
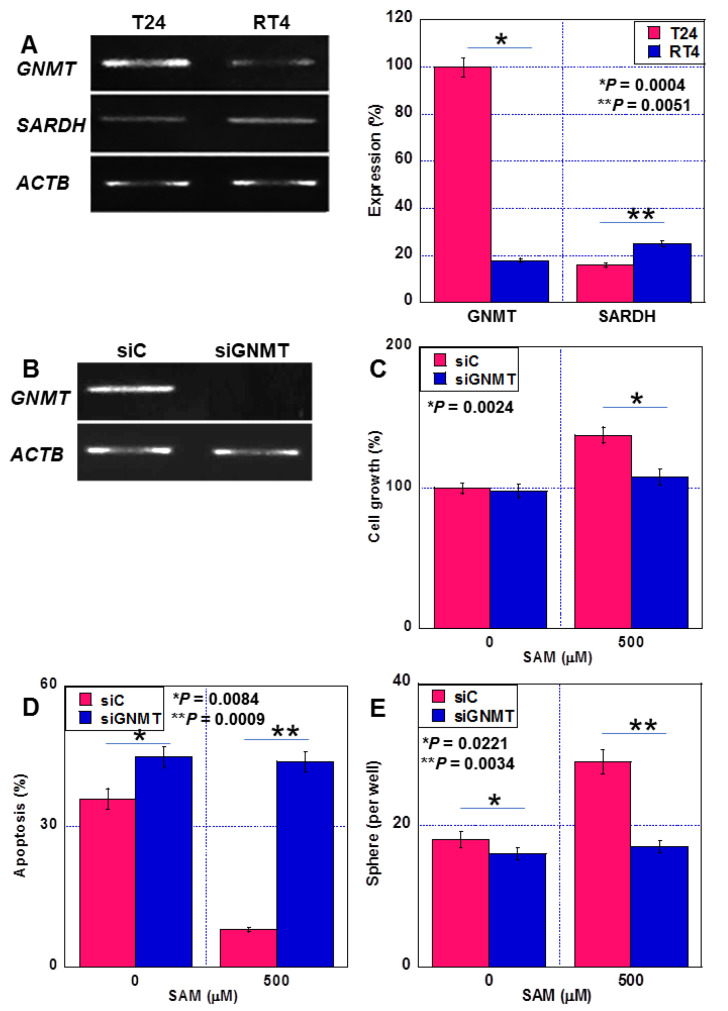
Effect of GNMT on human BUC cell malignancy. (**A**) Expression of GNMT and SADH in human BUC cells. RT-PCR image (left panel). Semi-quantitative data of RT-PCR (right panel). (**B**–**E**) Effects of GNMT knockdown on GNMT expression (**B**), cell growth (**C**), thapsigargin-induced apoptosis (**D**), and sphere formation (**E**). Error bars: standard deviation from three independent trials. Statistical significance was calculated using the Student’s *t*-test. The assumption test suggests that the difference between the two SDs is not significant. BUC, bladder urothelial carcinoma; SAM, S-adenosylmethionine; GNMT, glycine N-dehydrogenase; RT-PCR, reverse transcription- polymerase chain reaction; SARDH, sarcosine dehydrogenase; ACTB, β-actin; siC, control short hairpin RNA; siGNMT, short hairpin RNA to GNMT.

**Figure 4 ijms-24-16367-f004:**
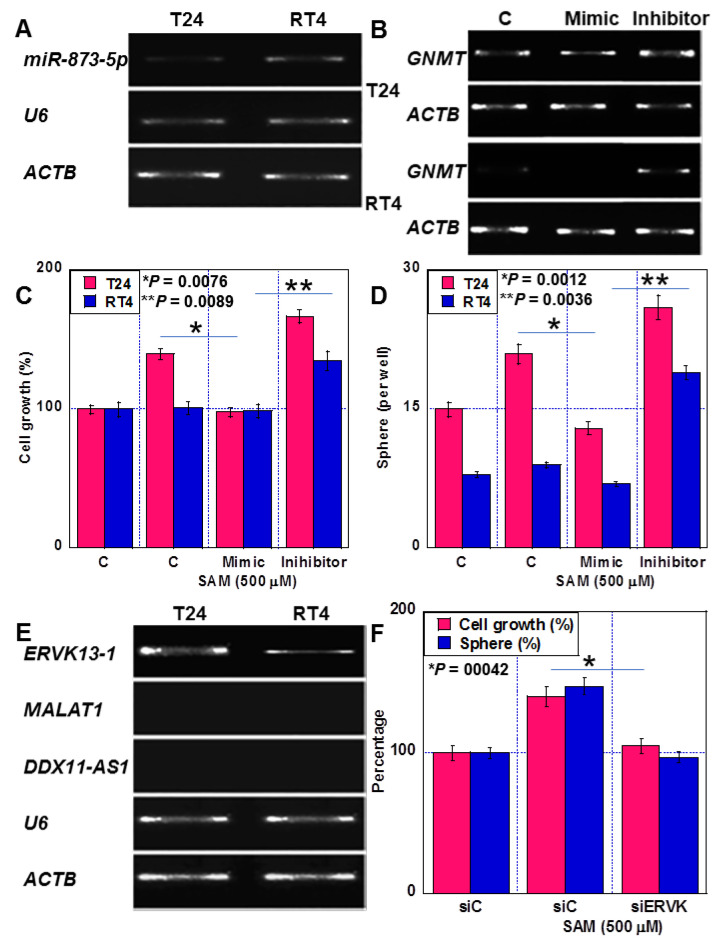
Epigenetic regulation of GNMT by miRNA and lncRNA in human BUC cells. (**A**) miR-873-5p expression (**B**–**D**) Effects of miR-873-5p mimic and miR-873-5p inhibitor on GNMT expression (**B**), cell growth (**C**), and sphere formation (**D**). (**E**) Expression of lncRNAs affecting miR-873-5p expression. (**F**) Effect of ERVK13-1 knockdown on cell growth and sphere formation. Error bars: standard deviation from three independent trials. Statistical significance was calculated using the Student’s *t*-test. The assumption test suggests that the difference between the two SDs is not significant. BUC, bladder urothelial carcinoma; SAM, S-adenosylmethionine; GNMT, glycine N-dehydrogenase; ACTB, β-actin; siC, control short hairpin RNA; siERVK, short hairpin RNA to ERVK13-1; ERVK13-1, endogenous retrovirus group K13 member 1; MALAT1, metastasis associated with lung adenocarcinoma transcript-1; DDX11-AS1, DDX11 antisense RNA 1; U6, U6 small nuclear RNA.

**Figure 5 ijms-24-16367-f005:**
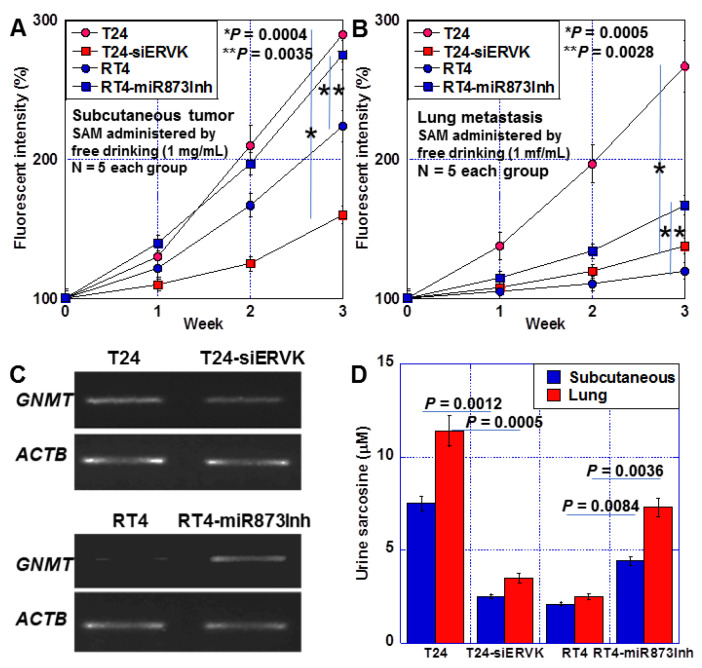
Effect of altered GNMT expression on tumor growth and metastability in human BUC cells. (**A**) Subcutaneous tumor volume by fluorescence intensity of GFP-transfected BUC cells pretreated with siERVK13-1 or miR-873-5p inhibitor before subcutaneous inoculation. (**B**) Lung metastatic tumor volume determined by fluorescence intensity of GFP-transfected BUC cells pretreated with siERVK13-1 or miR-873-5p inhibitor before caudal vein inoculation. (**C**) Expression of GNMT in subcutaneous tumors. (**D**) Urine sarcosine concentrations at euthanasia. Error bars: standard deviation from five mice. Statistical significance was calculated using the Student’s *t*-test. The assumption test suggests that the difference between the two SDs is not significant. BUC, bladder urothelial carcinoma; SAM, S-adenosylmethionine; GNMT, glycine N-dehydrogenase; ERVK13-1, endogenous retrovirus group K13 member 1; ACTB, β-actin; siC, control short hairpin RNA; siERVK, short hairpin RNA to ERVK13-1; miR-873Inh, miR-873-5p inhibitor.

**Figure 6 ijms-24-16367-f006:**
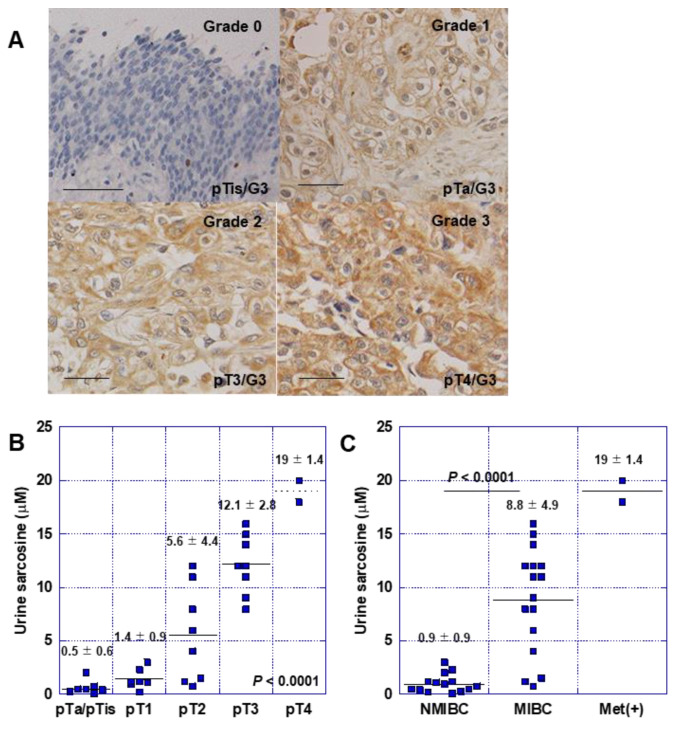
Role of GNMT in human BUC cases. (**A**) Immunostaining for GNMT in patients with BUC. Scale bar, 50 μm. (**B**,**C**) Association between urinary sarcosine concentration during surgery and tumor progression. Statistical significance was calculated using the Student’s *t*-test. Value is mean ± SD. BUC, bladder urothelial carcinoma; GNMT, glycine N-dehydrogenase; GNMT expression grade; NMIBC, non-invasive bladder cancer; MIBC, invasive bladder cancer.

**Figure 7 ijms-24-16367-f007:**
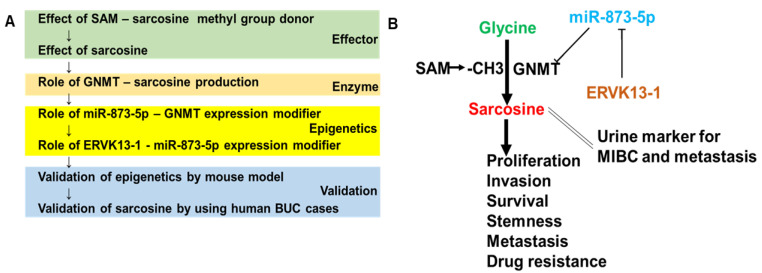
Sarcosine overproduction by ERVK13-1/miR-873-5p/GNMT axis. (**A**) Experimental design of this study. In order to clarify the role of sarcosine on BUC, we are proceeding with studies in the order of effector, enzyme, epigenetics, and validation. (**B**) Schematic summary of this study. Glycine receives a methyl group from SAM and is converted to sarcosine by GNMT. The expression of GNMT is suppressed by miR-873-5p, but the effect is lost due to lncRNA ERVK13-1, and GNMT is overexpressed. Sarcosine promotes BUC proliferation, invasion, survival, and stemness, leading to cancer metastasis and drug resistance. Urinary sarcosine can be a marker for MIBC and distant metastasis. BUC, bladder urothelial carcinoma; GNMT, glycine N-dehydrogenase; SAM, S-adenosylmethionine; MIBC, invasive bladder cancer; ERVK13-1, endogenous retrovirus group K13 member 1.

**Table 1 ijms-24-16367-t001:** GNMT expression in 86 patients with bladder urothelial carcinomas.

Parameter ^(1)^		n	GNMT Grade					*p* ^(3)^
			0	1	2	3	Positivity (%) ^(2)^	
Age	<60	43	18	12	5	8	30	
	>60	43	15	11	13	4	40	0.2369
Sex	M	71	28	17	17	9	37	
	F	15	5	6	1	3	27	0.3136
Tissue	Tumor	86	33	23	18	12	35	
	Normal	86	61	25	0	0	0	<0.0001
T factor	pTa/pTis	21	19	2	0	0	0	
	pT1	18	6	10	2	0	11	
	pT2	21	7	7	4	3	33	
	pT3	22	1	4	12	5	77	
	pT4	4	0	0	0	4	100	<0.0001
N factor	pN0	80	33	23	18	6	30	
	pN1-2	6	0	0	0	6	100	<0.0001
M factor	M0	82	33	23	18	8	32	
	M1b	4	0	0	0	4	100	<0.0001
pStage	0a/0is	21	19	2	0	0	0	
	I	18	6	10	2	0	11	
	II	21	7	7	4	3	33	
	IIIA	22	1	4	12	5	77	
	IVB	4	0	0	0	4	100	<0.0001
Muscle	NMIBC	39	25	12	2	0	5	
invasion	MIBU	47	8	11	16	12	60	<0.0001
Grade	G1	6	2	3	0	1	17	
	G2	33	20	7	5	1	18	
	G3	47	11	13	13	10	49	0.0132

^(1)^ TNM classification criteria were pT, pN, pM, stage, and grade [25]. pTa, non-invasive papillary urothelial carcinoma; pTis, urothelial carcinoma in situ; pT1, cancer grown into the layer of connective tissue under the lining layer; pT2, cancer grown into the muscle layer; pT3, cancer grown into the fatty tissue surrounding the bladder; pT4, cancer spread into the organs surrounding the bladder; pN0, no nodal metastasis; pN1-2, cancer spread to 1 or more lymph nodes in the true pelvis; M0, no spread to distant sites; M1b, spread to the lung; stage 0a/0is, pTa/pTis, pN0, M0; stage I, pT1, pN0, M0; stage II, pT2, pN0, M0; stage III, pT3, pN0, M0; stage IV, any pT, any pN, M1b. ^(2)^ Positivity: Since the normal urothelium showed grades 0 and 1, grades 2 and 3 were considered positive for GNMT. ^(3)^ Significant differences were calculated using the Chi-square test. GNMT, glycine N-methyltransferase; NMIBC, non-muscle invasive bladder cancer; MIBC, muscle-invasive bladder cancer.

**Table 2 ijms-24-16367-t002:** Primer sets.

Gene	Gene ID		Sequence
*CD44*	FJ216964.1	forward	AAGGTGGAGCAAACACAACC
		reverse	AGCTTTTTCTTCTGCCCACA
*OCT3*	BC117437.1	forward	GAAGGATGTGGTCCGAGTGT
		reverse	GTGAAGTGAGGGCTCCCATA
*GNMT*	KR710869.1	forward	CACCCCCAGGGAAGAACATC
		reverse	CCGTGAAGGATGCCAGACAG
*SARDH*	NM_007101.4	forward	GGAGGAGGAGACGGGACTAC
		reverse	CCGTAGAGGTCGTCCACATT
*ACTB*	NM_001101.3	forward	GGACTTCGAGCAAGAGATGG
		reverse	AGCACTGTGTTGGCGTACAG
*ERVK13-1*	NR_040023.1	forward	GATGTGCAGTGGGTGATGGA
		reverse	GCCAAGCCGCCTAATTCATG
*MALAT1*	NR_144568.1	forward	GGTTTCCCAGAGTCCTTGGG
		reverse	TCAATCCCACACCACAGAGC
*DDX11-AS1*	NR_038927.2	forward	CGATTAGCGCCAGGTGTACT
		reverse	AAAGGTTGCTGGCTGATGGT
*miR-873-5p*	NR_030618.1	forward	TGCAGGAACTTGTGAGTCTCC
		reverse	TTCCCGGGAACTCATCAGTC
*U6*	EU520423	forward	TTATGGGTCCTAGCCTGAC
	[38]	reverse	CACTATTGCGGGTCTGC

OCT3, octamer binding transcription factor; GNMT, glycine N-dehydrogenase; SARDH, sarcosine dehydrogenase; ACTB, β-actin; ERVK13-1, endogenous retrovirus group K13 member 1; MALAT1, metastasis-associated lung adenocarcinoma transcript-1; DDX11-AS1, DDX11 antisense RNA 1; U6, U6 small nuclear RNA.

## Data Availability

Data are contained within the article.

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
