# Peer review of "ERVK13-1/miR-873-5p/GNMT Axis Promotes Metastatic Potential in Human Bladder Cancer though Sarcosine Production"

_ijms, 2023, doi:10.3390/ijms242216367_

Round 1
Reviewer 1 Report
Comments and Suggestions for Authors
Estimated Authors,
I've been invited to review the present paper entitled "ERVK13-1/miR-873-5p/GNMT axis promotes metastatic potential in human bladder cancer though sarcosine production". In this study, Kishi et al. were able to demonstrate that sarcosine levels (sarcosine is produced by glycine N-methyltransferase (GNMT) by transferring 20 a methyl group from S-adenosyl methionine (SAM) to glycine) were strongly associated with malignant phenotypes in patients with BUC.
Moreover, the external administration of sarcosine enhanced the proliferation (?) of both T24 cells, which overexpress GNMT, and RT4 cells.
The present study is substantially well written, and I've detected no significant flaws. On the other hand, I've some concerns about the graphs and images associated with the study.
For example, Figure 1D, Figure 1G, Figure 3A + B, Figure 4 A, B, E, are of low quality (in my copy: it could be the consequence of the *.docx to *.pdf conversion). I would suggest to include a high resolution copy as annex material.
Regarding the other figures, I would remove dotted lines in order to improve the readability.
Author Response
Responses to reviewer’s comments
Reviewer 1
Estimated Authors,
I've been invited to review the present paper entitled "ERVK13-1/miR-873-5p/GNMT axis promotes metastatic potential in human bladder cancer though sarcosine production". In this study, Kishi et al. were able to demonstrate that sarcosine levels (sarcosine is produced by glycine N-methyltransferase (GNMT) by transferring 20 a methyl group from S-adenosyl methionine (SAM) to glycine) were strongly associated with malignant phenotypes in patients with BUC.
Moreover, the external administration of sarcosine enhanced the proliferation (?) of both T24 cells, which overexpress GNMT, and RT4 cells.
The present study is substantially well written, and I've detected no significant flaws. On the other hand, I've some concerns about the graphs and images associated with the study.
For example, Figure 1D, Figure 1G, Figure 3A + B, Figure 4 A, B, E, are of low quality (in my copy: it could be the consequence of the *.docx to *.pdf conversion). I would suggest to include a high resolution copy as annex material.
Regarding the other figures, I would remove dotted lines in order to improve the readability.
(Answer) Thank you for your positive comments. The images in your question were replaced with higher-resolution, sharper ones. We also created graphs without grids using dotted lines, but many people commented that they were difficult to understand compared to graphs with grids. In addition, we have presented data using graphs with grids and published papers without any problems. Therefore, I decided to use a graph with a grid again this time.
Reviewer 2 Report
Comments and Suggestions for Authors
Despite the extensive experimental work by the authors of this manuscript, the manuscript presentation needs to be more organized. Besides, several concerns to be addressed, and some editing is required as follows:
1. The abstract word count is too high. Please adhere to word limits in instructions to authors.
2. The introduction needs to be fortified with a more detailed background on sarcosine.
3. Results:
• The exact P-value needs to be provided in the result section, for a clear understanding of statistical significance.
• In all figure legends and table footnotes: the full term of all abbreviations used should be clarified. Clarify the number of replicates n=?
• Ordinary ANOVA is a general term. Specify one way or two way.
• Figure 1: give a symbol “D” for the representative image. Additionally, the details of the image are unclear and need to be replaced with a better one. Moreover, the figure legend, describes what exists in the image.
4. The discussion needs to be deepened by more interpretations of the study findings and correlate the relation of the estimated parameters rather than just mentioning the results. Also, the authors should clarify the limitations of the study.
5. Material and methods:
- The authors are highly recommended to add a schematic figure summarizing the experiments performed and the estimated parameters.
In line 296, the authors mentioned: “To assess human and murine mRNA expression”. However, in Table 2, the authors presented only Human genes. Also, remove the animal from the top row.
- Line 329: why did the authors add SAM in drinking water and not administer via oral gavage to control the dosage volume? Also, on what basis did the authors choose the dose of 6.6 mg/mouse/day? It is better to present the doses as mg/kg and justify the dose selection with references.
- The ethical justification for the experiment and animal welfare details have not been clarified. What is the initial average weight of the mice used? How the mice were euthanatized? Has the mice were anesthetized?
- Line 341: add the reference to the TRIzol method.
- The authors depend on secondary, not original, references in several paragraphs and methods. For instance, line 353 Immunohistochemistry, [42] Fujiwara-Tani et al., 2023 is not the original reference for the immunohistochemistry methods. This is an important issue that needs to be revised and fixed throughout the manuscript.
- Reference [43] is not present in the text.
- Statistical analysis: many details are missed as follows:
• Have the data been presented as means±SE or SD?
• What is the software used?
• Does data meet the assumption of homogeneity of variances and normal distribution? Clarify if the authors run a homogeneity or normality test.
• The type of ANOVA used and for what parameters.
Comments on the Quality of English LanguageGood language.
Author Response
Responses to reviewer’s comments
Reviewer 2
Despite the extensive experimental work by the authors of this manuscript, the manuscript presentation needs to be more organized. Besides, several concerns to be addressed, and some editing is required as follows:
- The abstract word count is too high. Please adhere to word limits in instructions to authors.
(Answer) The word count for the abstract is 298 words, which we believe is within the limit. Words that seemed abstract were deleted or rewritten.
- The introduction needs to be fortified with a more detailed background on sarcosine.
(Answer) Added description of sarcosine dehydrogenase, a sarcosine decomposition system. This enzyme is also mentioned in Figure 3 and seems to be significant.
- Results:
- The exact P-value needs to be provided in the result section, for a clear understanding of statistical significance.
(Answer) All P values are listed in the figures.
- In all figure legends and table footnotes: the full term of all abbreviations used should be clarified. Clarify the number of replicates n=?
(Answer) The n value and the spelling out of the abbreviated word are listed in the legend.
- Ordinary ANOVA is a general term. Specify one way or two way.
(Answer) Added that the ANOVA test is two-way.
- Figure 1: give a symbol “D” for the representative image. Additionally, the details of the image are unclear and need to be replaced with a better one. Moreover, the figure legend, describes what exists in the image.
(Answer) Added D in the figure. We added explanation of panel D.
- The discussion needs to be deepened by more interpretations of the study findings and correlate the relation of the estimated parameters rather than just mentioning the results. Also, the authors should clarify the limitations of the study.
(Answer) Following your suggestion, we created a scheme that integrates the data from this study (Figure 7) and created a new Conclusion section based on it. We added the limitations regarding our research to the discussion.
- Material and methods:
- The authors are highly recommended to add a schematic figure summarizing the experiments performed and the estimated parameters.
(Answer) A schematic diagram was created and presented as Figure 7.
In line 296, the authors mentioned: “To assess human and murine mRNA expression”. However, in Table 2, the authors presented only Human genes. Also, remove the animal from the top row.
(Answer) The animal column has been deleted.
- Line 329: why did the authors add SAM in drinking water and not administer via oral gavage to control the dosage volume? Also, on what basis did the authors choose the dose of 6.6 mg/mouse/day? It is better to present the doses as mg/kg and justify the dose selection with references.
(Answer) The concentration of SAM was set at a daily dose of 20 g/kgBW/day. The values listed are based on the average amount of water consumed by mice. The reason for providing free drinking water was because it was thought that the tumor could be exposed to a more average concentration. We also added a reference.
- The ethical justification for the experiment and animal welfare details have not been clarified. What is the initial average weight of the mice used? How the mice were euthanatized? Has the mice were anesthetized?
(Answer) We added the initial weight of mice in Methods. We also described that euthanasia was performed under inhalation anesthesia.
- Line 341: add the reference to the TRIzol method.
(Answer) We cited the literature.
- The authors depend on secondary, not original, references in several paragraphs and methods. For instance, line 353 Immunohistochemistry, [42] Fujiwara-Tani et al., 2023 is not the original reference for the immunohistochemistry methods. This is an important issue that needs to be revised and fixed throughout the manuscript.
(Answer) Many experimental techniques remain essentially the same, but are gradually modified and require new references. Therefore, we are quoting new papers from our research group. However, we followed your instructions and cited the original document.
- Reference [43] is not present in the text.
(Answer) Reference “43” is shown in Table 1 as “40” in the revised manuscript.
- Statistical analysis: many are missed as follows:
- Have the data been presented as means±SE or SD?
- What is the software used?
- Does data meet the assumption of homogeneity of variances and normal distribution? Clarify if the authors run a homogeneity or normality test.
- The type of ANOVA used and for what parameters.
(Answer)
・The legend indicates that the values in the figure are mean ± SD.
・The software is described in Methods.
・Mann-Whitney U test was performed for those that did not satisfy normality or were shown to not meet normality.
・It was clearly stated that it was a two-way ANOVA.
Round 2
Reviewer 2 Report
Comments and Suggestions for Authors
The authors need to address the comments more carefully as many of the previous comments have not been addressed as follows:
- According to the instructions of the journal: the abstract should be a single paragraph of about 200 words maximum. However, the authors believe that a 298-word count is appropriate.
- The background on sarcosine is still very concise and has not been adequately revised.
- For a clear understanding of statistical significance, the exact P-value needs to be provided in the result section, not just mentioning p<0.05 in the figures as the authors mentioned.
- In all figure legends and table footnotes: the full term of all abbreviations used should be clarified. Clarify the number of replicates n=?
- (D) Microscopic image of the sphere. Complete the illustration.
- The discussion needs to be deepened by more interpretations of the study findings and correlate the relation of the estimated parameters rather than just mentioning the results.
- The authors have added Figure 7 in response to this comment (The authors are highly recommended to add a schematic figure summarizing the experiments performed and the estimated parameters). However, this figure is a schematic presentation of the results of the study, not the experimental design.
- Line 346: clarify the exact type of inhalation anesthesia used.
- The authors mentioned that (the Mann-Whitney U test was performed for those that did not satisfy normality or were shown to not meet normality). How the normality was checked?
Comments on the Quality of English Language-
Author Response
Responses to the reviewer’s comments
The authors need to address the comments more carefully as many of the previous comments have not been addressed as follows:
- According to the instructions of the journal: the abstract should be a single paragraph of about 200 words maximum. However, the authors believe that a 298-word count is appropriate.
(Answer) We shortened Abstract to 188 words.
- The background on sarcosine is still very concise and has not been adequately revised.
(Answer) We added sentence explanation of sarcosine to be associated with aging and autophagy.
- For a clear understanding of statistical significance, the exact P-value needs to be provided in the result section, not just mentioning p<0.05 in the figures as the authors mentioned.
(Answer) We described exact P values in figures.
- In all figure legends and table footnotes: the full term of all abbreviations used should be clarified. Clarify the number of replicates n=?
(Answer) The full term of all abbreviations and the number of replicates of experiments were designated in figure legends and table footnote.
- (D) Microscopic image of the sphere. Complete the illustration.
(Answer) We added explanation of the image in the figure.
- The discussion needs to be deepened by more interpretations of the study findings and correlate the relation of the estimated parameters rather than just mentioning the results.
(Answer) Using Fig. 7A, which we created to answer the following questions, we added a text that provides an overview of this research to the discussion.
- The authors have added Figure 7 in response to this comment (The authors are highly recommended to add a schematic figure summarizing the experiments performed and the estimated parameters). However, this figure is a schematic presentation of the results of the study, not the experimental design.
(Answer) We added a diagram summarizing the experimental design in Figure 7 and explained it at the beginning of the discussion.
- Line 346: clarify the exact type of inhalation anesthesia used.
(Answer) We used isoflurane, which added in Methods.
- The authors mentioned that (the Mann-Whitney U test was performed for those that did not satisfy normality or were shown to not meet normality). How the normality was checked?
(Answer) By using InStat software, Assumption test suggests that the difference between the two SDs is not significant. Then we recalculate P values by Student’s t-test.
Round 3
Reviewer 2 Report
Comments and Suggestions for Authors
-